# A Novel Memristive Neural Network Circuit and Its Application in Character Recognition

**DOI:** 10.3390/mi13122074

**Published:** 2022-11-25

**Authors:** Xinrui Zhang, Xiaoyuan Wang, Zhenyu Ge, Zhilong Li, Mingyang Wu, Shekharsuman Borah

**Affiliations:** 1School of Electronics and Information, Hangzhou Dianzi University, Hangzhou 310018, China; 2Department of Electronics and Communication Engineering, Indian Institute of Information Technology, Guwahati 781015, India

**Keywords:** artificial neural network (ANN), character picture recognition, memristor, memristive neural network (MNN), synaptic circuit, neural network circuit

## Abstract

The memristor-based neural network configuration is a promising approach to realizing artificial neural networks (ANNs) at the hardware level. The memristors can effectively simulate the strength of synaptic connections between neurons in neural networks due to their diverse significant characteristics such as nonvolatility, nanoscale dimensions, and variable conductance. This work presents a new synaptic circuit based on memristors and Complementary Metal Oxide Semiconductor(CMOS), which can realize the adjustment of positive, negative, and zero synaptic weights using only one control signal. The relationship between synaptic weights and the duration of control signals is also explained in detail. Accordingly, Widrow–Hoff algorithm-based memristive neural network (MNN) circuits are proposed to solve the recognition of three types of character pictures. The functionality of the proposed configurations is verified using SPICE simulation.

## 1. Introduction

Currently, researchers are giving considerable attention to the implementation of neural networks on hardware platforms to enhance data processing efficiency. The memristive neural network (MNN) circuit is a hardware system that can incorporate memory and computation. It is suitable for high-speed parallel computation and solving the efficiency issues driven by the bottleneck of Von Neumann. Thus, the MNN circuit is a potential candidate for the realization of ANN [1,2,3,4,5,6,7,8,9]. The unique nonvolatile attributes of memristors and synapses are quite comparable in terms of memory characteristics [10]. To express the weight of synapses, memristors can be directly utilized, which further recognize the application of memristors in neural networks.

The essential link in MNN circuit design is the design of the synaptic circuit [11,12,13,14,15,16,17]. Four memristor-based synaptic bridge circuits, which can realize positive, zero, and negative weights is reported in [18,19,20]. In [20,21], it is suggested to use two cross arrays with the same structure, in which two memristors in the same position act as synaptic circuits. The same input signal is applied to two memristor crossed arrays to obtain the output, and then the difference between the memristors is mapped to positive, zero, or negative weights. Then, the neural network circuit based on the memristors is used to realize character recognition. In [22,23,24], differential input signals were applied to two rows of a memristor cross array. The sum of the output voltages was expressed as the difference between the resistance values of two memristors, thus obtaining positive, zero, and negative weights. However, when using the array as a synaptic circuit, selecting a certain row or column of the array is necessary, which leads to the inability to realize parallel programming of synaptic circuits in the whole network during operation, which limits the development of accelerated calculation of neural network. In [25], four Metal-Oxide-Semiconductor (MOS) transistors and a complementary resistance switch were used to form a memory cell. Only positive voltage was used to adjust the resistance, thus simplifying the power supply design and making the control circuit easier to realize. However, this circuit needs two kinds of control signals to adjust the memristance. A 1T2M (one MOS transistor and two memristors) structure memristive synapse circuit was reported in [26]. Specifically, the MOS transistor was used as a switch to determine whether the circuit updated the weight or saved the weight, but the circuit could only realize the positive weight. In [27], a 4T2M (four MOS transistors and two memristors) structure memristive synapse circuit was designed, which required two different control voltages to control the weights in the circuit. In [28], a 4T1M (four MOS transistors, one memristor, and an inverter) structure memristor synapse circuit was designed, which required two kinds of control voltage signals to be applied to the control terminal of the circuit at the same time through an inverter. According to the characteristics of the above circuits, this paper optimizes the above circuit structures and reduces the number of control voltages.

In this paper, a simple memristive synapse circuit is implemented which can adjust the positive, negative, and zero synapse weights through only one control terminal. The memristive neuron circuit is realized by designing a signal summation and activation function circuit. Lastly, through combination with the Widrow–Hoff algorithm, a single-layer neural network circuit is designed to recognize three types of character pictures.

The remainder of this paper is organized as follows: Section 2 presents the design of the memristive synapse circuit along with the description of the circuit weighting operation and weight programming operation; Section 3 presents the design of the neuron circuit, as well as the simulation outcomes ascertaining the function of signal summation and activation; Section 4 shows the character recognition network circuit and confirms the circuit recognition’s accuracy using software; lastly, Section 5 summarizes the paper.

## 2. Design of Memristive Synapse Circuit

### 2.1. Memristor Model

The HP memristor is a continuous memristor whose resistance changes continuously under the action of applied voltage. It has the advantages of nonvolatility and nanoscale operation, which make it suitable for the design of MNN circuits [3]. The simplified mathematical model of the HP memristor can be expressed as
(1)$$M(q)=R_{\mathrm{ON}}\frac{\mu_{v}R_{\mathrm{ON}}}{D^{2}}q(t)+R_{\mathrm{OFF}}(1-\frac{\mu_{v}R_{\mathrm{ON}}}{D^{2}}q(t))$$
where *M*(*q*) represents the memresistance (Ω), *q*(*t*) represents the amount of charge (C) flowing through the memristor, *R*_ON_ and *R*_OFF_ represents the resistance of the doped and undoped region, *D* represents the length (nm) of TiO_2_, and *μ* represents the average ion mobility (cm^2^/Vs) of the material.

### 2.2. Weighted Operation

Figure 1a presents a new memristive synaptic circuit by applying the HP memristor model. The circuit is composed of four MOS transistors (T_1_–T_4_) and two memristors (*M*_A_ and *M*_B_) connected in reverse series. Here, T_1_ and T_4_ are the PMOS transistors, T_2_ and T_3_ are NMOS transistors, and *V*_in_ is the input of the synaptic circuit. The control voltage *V*_G_ acts on the gates of the four MOS transistors simultaneously to adjust the positive and negative sign of the weight.

The output voltage, *V*_out_ is the voltage difference between nodes *A* and *B* under the given input, i.e., *V*_out_ = *V*_A_ − *V*_B_. Because the two memristors in this circuit are identical and connected in reverse series, the resistance changes of *M*_A_ and *M*_B_ are always opposite; hence, their sum remains a constant value, i.e., *M*_A_ + *M*_B_ = *R*_ON_ + *R*_OFF_. The sign of the synaptic circuit’s weight can be specified by the control voltage, *V*_G_, as shown in Table 1.

As shown in Table 1 and Figure 1b, *Condition 1* depicts the current path in the circuit. The direction of the current flowing through the memristor *M*_B_ is from node A to node B. The output voltage *V*_out_ is greater than 0 and can be expressed as
(2)$$V_{\mathrm{out}}=V_{A}-V_{B}=\frac{M_{B}}{M_{A}+M_{B}}V_{\mathrm{in}}$$

Likewise, as shown in Table 1 and Figure 1c, *Condition 2* affirms that the current of the memristor *M*_B_ flows from node B to node A, and the output voltage *V*_out_ is negative, as shown below.
(3)$$V_{\mathrm{out}}=V_{A}-V_{B}=-\frac{M_{B}}{M_{A}+M_{B}}V_{\mathrm{in}}$$

Therefore, the following relationship can be observed with respect to the input and output voltages of the synaptic circuit:(4)$$V_{\mathrm{out}}=\omega V_{\mathrm{in}}=\frac{V_{G}}{\left|V_{G}\right|}\cdot \frac{M_{B}}{M_{A}+M_{B}}V_{\mathrm{in}}$$
where $\omega =\frac{V_{G}}{\left|V_{G}\right|}\cdot \frac{M_{B}}{M_{A}+M_{B}}$, for *R*_ON_ << *R*_OFF_, and the memristances *M*_A_ and *M*_B_ are all within [*R*_ON_, *R*_OFF_]; thus, when *M*_B_ = *R*_ON_, *ω =* ±*R*_ON_/(*R*_OFF_ + *R*_ON_) ≈ 0, and, when *M*_B_ = *R*_OFF_, *ω =* ± *R*_OFF_/(*R*_ON_ + *R*_OFF_) ≈ 1. It can be concluded that *ω* can be changed in the range of [−1, 1]. Therefore, *ω* can be used to represent the weight of synaptic circuits and can realize “positive”, “negative”, and “zero” weights.

### 2.3. Weight Programming Operation

According to the circuit structure and the working mechanism of memristors, the relationship between the weight change of the synaptic circuit and the action time *t* of the programming voltage can be analyzed. Concretely, because the two memristors are connected in reverse series, when the programming voltage *V*_p_ = + 5 V is applied to the input of the synaptic circuit, the changes in *M*_A_(*t*) and *M*_B_(*t*) are opposite, resulting in the total memristance *M*(*t*) = *M*_A_(*t*) + *M*_B_(*t*) in the circuit remaining unchanged. Let *M*_A_(0) and *M*_B_(0) represent the initial values of the two memristors; combined with the memristor model in Equation (1) and the control voltage *V*_G_, the memristances of *M*_A_ and *M*_B_ in the weight programming stage can be obtained as follows:(5)$$\{\begin{array}{c} M_{A}(t)=M_{A}(0)-(R_{\mathrm{OFF}}-R_{\mathrm{ON}})\times k\times q_{1}(t)\\ M_{B}(t)=M_{B}(0)+(R_{\mathrm{OFF}}-R_{\mathrm{ON}})\times k\times q_{2}(t) \end{array}\begin{array}{cc} & V_{G}>0 \end{array}$$
(6)$$\{\begin{array}{c} M_{A}(t)=M_{A}(0)+(R_{\mathrm{OFF}}-R_{\mathrm{ON}})\times k\times q_{1}(t)\\ M_{B}(t)=M_{B}\left(0\right)-(R_{\mathrm{OFF}}-R_{\mathrm{ON}})\times k\times q_{2}(t) \end{array}\begin{array}{cc} & V_{G}<0 \end{array}$$
where *R*_OFF_ represents the high-resistance state of the memristor, and *R*_ON_ represents the low-resistance state. *k* = *μ_v_* × *R*_ON_/*D*^2^ is a constant. Since the amount of charge flowing through the two memristors in the series circuit is always the same, i.e., *q*_1_(*t*) = *q*_2_(*t*), the total memristance *M*(*t*) of the synaptic circuit can be further expressed as
(7)$$M(t)=M_{A}(t)+M_{B}(t)=M_{A}(0)+M_{B}(0)$$
when *V*_G_ > 0, according to the current flow direction in Figure 1b and Equations (5) and (7), the corresponding weight change can be obtained as follows:(8)$$\Delta \left|\omega (t)\right|=\frac{\Delta M_{B}(t)}{M_{A}(t)+M_{B}(t)}=-\frac{k\times (R_{\mathrm{OFF}}-R_{\mathrm{ON}})\times \Delta q(t)}{M_{A}(0)+M_{B}(0)}=A\times \Delta t>0$$
where *A* = *k* × (*R*_OFF_ − *R*_ON_) × *I*/(*M*_A_(0) + *M*_B_(0)) = 30.67 can be obtained by substitute the parameters into the formula, which is a fixed value. As the resistance of *M*_B_ is within the range [100, 16K], the maximum range of ∆*M*_B_ is 15.9 kΩ; thus, ∆ω = 15.9 kΩ/16.1 kΩ ≈ 0.988. According to Equation (8), the range of ∆*t* can be obtained as [0, 0.032].

Similarly, when *V*_G_ < 0, the weight change can be obtained as follows:(9)$$\Delta \left|\omega (t)\right|=\frac{\Delta M_{B}(t)}{M_{A}(t)+M_{B}(t)}=-\frac{k\times (R_{\mathrm{OFF}}-R_{\mathrm{ON}})\times \Delta q(t)}{M_{A}(0)+M_{B}(0)}=-A\times \Delta t<0$$

According to the above analysis, the weight change of the synaptic circuit at any time in the weight programming stage can be expressed as
(10)$$\Delta \left|\omega (t)\right|=\frac{V_{G}}{\left|V_{G}\right|}\times \frac{\Delta M_{B}(t)}{M_{A}(t)+M_{B}(t)}=\frac{V_{G}}{\left|V_{G}\right|}\times \frac{k\times (R_{\mathrm{OFF}}-R_{\mathrm{ON}})\times \Delta q(t)}{M_{A}(0)+M_{B}(0)}=A\times \frac{V_{G}}{\left|V_{G}\right|}\times \Delta t$$

Thus, the weight *ω*(*t*) of the synaptic circuit at any moment can be obtained as follows:(11)$$\omega (t)=\frac{V_{G}}{\left|V_{G}\right|}[\left|\omega (0)\right|+\Delta \left|\omega (t)\right|]=\frac{V_{G}}{\left|V_{G}\right|}\times \left|\omega (0)\right|+A\times \Delta t$$
where $\omega (0)=\frac{V_{G}}{\left|V_{G}\right|}\times \frac{M_{B}(0)}{M_{A}(0)+M_{B}(0)}$.

Equation (11) presents the linear functional relationship between the synaptic circuit’s weight *ω*(*t*) and the action time *t* of the programming voltage *V*_p_ at any moment. According to the positive and negative control voltage *V*_G_, the operation of increasing or decreasing the weight of the synaptic circuit can be realized.

A comparative analysis of the functions of the proposed memristor synaptic circuit with previously reported studies [26,27,28,29,30] is presented in Table 2. The proposed synaptic circuit offers various advantages in terms of the number of control voltages and the weight range, and it provides good linearity in the programming stage. Therefore, the proposed configuration has better operability in the weight programming stage.

## 3. The Neuron Circuit

### 3.1. The Neuron Circuit Design

The synaptic and neuron circuits are two basic units in the MNN circuit. An example of the proposed configuration is verified by considering a neuron circuit with two connected synaptic circuits as shown in Figure 2. The left dashed box represents the two memristive synaptic circuits designed in Section 2, and the right dashed box represents the neuron circuit. The potential difference between *A_i_* and *B_i_* (*I* = 1, 2) is obtained using the two subtractors composed of operational amplifiers A1 and A2, and resistors *R*_1–8_. Specifically, when *R*_1_–*R*_8_ are equal, *V*_O1_ = *V*_A1_ − *V*_B1_ = *ω*_1_×*V*_in1_ and *V*_O2_ = *V*_A2_ − *V*_B2_ = *ω*_2_ × *V*_in2_. The operational amplifier A3 and resistor *R*_9–13_ constitute an in-phase addition circuit. The output voltage is *V*_O_ = *V*_O1_ + *V*_O2_ = *ω*_1_ × *V*_in1_ + *ω*_2_ × *V*_in2_, when *R*_12_ = *R*_9_ = *R*_10_ = *R*_11_ and *R*_6_ = *R*_9_//*R*_10_.

Similarly, when a neuron circuit is connected to *n* synaptic circuits, the relationship between its input and output can be expressed as
(12)$$V_{O}=\sum_{i=1}^{n}\frac{V_{Gi}}{\left|V_{Gi}\right|}\times \frac{M_{B}{}_{i}(t)}{M_{A}{}_{i}(t)+M_{B}{}_{i}(t)}\times V_{\mathrm{in}i}=\sum_{i=1}^{n}\omega_{i}(t)\times V_{\mathrm{in}i}$$

It can be seen that the neuron circuit realizes the weighted summation of input signals. In Figure 2, the activation function output of the neuron circuit composed of NMOS transistor T_9_ and resistor *R*_14_ is as follows:(13)$$sign(V_{O})=\{\begin{array}{c} 1,\begin{array}{ccc} & & \mathrm{if}\; \end{array}V_{O}\geq 0\\ 0,\begin{array}{ccc} & & \mathrm{if}\; \end{array}V_{O}<0 \end{array}$$

### 3.2. Simulation Analysis

In the simulation, all the operational amplifiers and MOSFETs are chosen as the universal ones. By considering *R_i_* = 100 kΩ (*I* = 1, …, 12), *R*_13_ = 50 kΩ, *R*_14_ = 10 kΩ, *V*_G1_ = 5 V, *V*_G2_ = −5 V and the values of HP memristors *R*_ON_, *R*_OFF_, *D*, and *μ_v_* were set to 100 Ω, 16 kΩ, 10 nm, and 10^−13^ m^2^·s^−1^·V^−1^, respectively. The initial values of *M*_A1_ and *M*_B2_ are 16 kΩ. According to $\omega =\frac{V_{G}}{\left|V_{G}\right|}\times \frac{M_{B}}{M_{A}+M_{B}}$, it can be known that the initial values of weights at this time are *ω*_1_(0) ≈ 0 and *ω*_2_(0) ≈ −1, respectively. Figure 3 shows the simulation results of the LTSpice neuron circuit with two synaptic circuits, where *V*_p1,2_ are weighted programming voltages applied to the inputs of two synaptic circuits in the programming stage. *I*_1_ and *I*_2_ are the currents of the memristor *M*_B*i*_ (*i* = 1, 2) flowing from the negative electrode to the positive electrode in the two synaptic circuits.

In the time period 0–10 ms, *V*_p1,2_ = *V*_in1_ = *V*_in2_ = 0 V, while the memristance and the weights *ω*_1_(0) ≈ 0 and *ω*_2_(0) ≈ −1 remain unchanged.

The first weight programming stage initiates at 10–40 ms, and the weight programming voltage *V*_p1,2_ = 5 V. According to the initial values of the memristors, since the control voltage *V*_G1_ of synaptic circuit 1 is kept at 5 V and the initial weight *ω*_1_(0) ≈ 0, the initial value of current *I*_1_ can be calculated as 0.31 mA. Likewise, the initial value of current *I*_2_ is −0.31 mA with *V*_G2_ = −5 V, *ω*_2_(0) ≈ −1. In this process, according to the initial values of *M*_A1_ and *M*_B2_ being 16 kΩ, *M*_A2_ and *M*_B1_ are 100 Ω; combined with Equations (5) and (6), the instantaneous resistances of the memristors at each moment can be obtained as follows:(14)$$\left\{ \begin{array}{c} M_{A1}(t)=M_{B2}(t)=16K-(16K-100)\times 10^{5}\times 0.00031(t-0.01)\\ =16K-4.929\times 10^{5}(t-0.01)(\Omega )\\ M_{B1}(t)=M_{A2}(t)=100+(16K-100)\times 10^{5}\times 0.00031(t-0.01)\\ =100+4.929\times 10^{5}(t-0.01)(\Omega ) \end{array}\right.$$

At the same time, according to Equations (10) and (11), the weight changes of the two synaptic circuits and the weights at any time can be obtained as follows:(15)$$\begin{array}{c} \Delta \left|\omega_{1}(t)\right|=\frac{\Delta M_{B1}(t)}{M_{A1}(t)+M_{B1}(t)}=\frac{k\times (R_{\mathrm{OFF}}-R_{\mathrm{ON}})\times I_{1}(t-0.01)}{M_{A1}(0)+M_{B1}(0)}\\ =30.615(t-0.01)>0\\ \Delta \left|\omega_{2}(t)\right|=\frac{\Delta M_{B2}(t)}{M_{A2}(t)+M_{B2}(t)}=-\frac{k\times (R_{\mathrm{OFF}}-R_{\mathrm{ON}})\times I_{2}(t-0.01)}{M_{A2}(0)+M_{B2}(0)}\\ =-30.615(t-0.01)<0 \end{array}$$
(16)$$\begin{array}{c} \omega_{1}(t)=\frac{V_{G1}}{\left|V_{G1}\right|}[\left|\omega_{1}(0)\right|+\Delta \left|\omega_{1}(0.03)\right|]=\frac{V_{G1}}{\left|V_{G1}\right|}[\left|0\right|+30.615(t-0.01)]=30.615(t-0.01)\\ \omega_{2}(t)=\frac{V_{G2}}{\left|V_{G2}\right|}[\left|\omega_{2}(0)\right|+\Delta \left|\omega_{2}(0.03)\right|]=\frac{V_{G2}}{\left|V_{G2}\right|}[\left|-1\right|-30.615(t-0.01)]=30.615(t-0.01)-1 \end{array}$$

Furthermore, the sum of weighted signals *V*_O_ and the output voltage *V*_out_ of the neuron circuit at each time can be derived as follows:(17)$$V_{O}=\omega_{1}\times V_{\mathrm{in}1}+\omega_{2}\times V_{\mathrm{in}2}=306.15(t-0.01)-5$$
(18)$$V_{\mathrm{out}}=sign(V_{O})=sign(306.15(t-0.01)-5)=\{\begin{array}{c} 1,\begin{array}{ccc} & & \mathrm{if}\; \end{array}t\geq 26.33\;\mathrm{ms}\\ 0,\begin{array}{ccc} & & \mathrm{if}\; \end{array}t<26.33\;\mathrm{ms} \end{array}$$

Again, *V*_G1_ changed from +5 V to −5 V, while *V*_G2_ changed from −5 V to +5 V in the time period 40–45 ms, and the current path was set in advance for the next weight programming stage. However, since the weight programming voltage *V*_p1,2_ = 0 V at this stage, the resistance of each memristor and the weight *ω_i_* (*i* = 1, 2) in the synaptic circuit remain unchanged, and *V*_out_ = 0 V.

The second weight programming phase starts at 45–75 ms, and the weight programming voltage is *V*_p1,2_ = 5 V. At this time, according to Equation (14), the initial value of the memristor is *M*_B2_ (0.045) = *M*_A1_ (0.045) = *M*_A1_ (0.04) = 1213 Ω, *M*_B1_ (0.045) = *M*_A2_ (0.045) = *M*_A2_ (0.04) = 14,887 Ω. The control voltage *V*_G1_ of synapse circuit 1 is at a low level, with the initial weight *ω*_1_ (0.045) = −0.918, the control voltage *V*_G2_ of synapse circuit 2 is at a high level, with the initial weight *ω*_2_ (0.045) = +0.082, and the current *I*_1_ and the current *I*_2_ are −0.31 mA and 0.31 mA, respectively, at 45 ms. The resistances of the memristors are *M*_A_(*t*) = *M*_B2_(*t*) =1213 − 4.929 × 10^5^(*t* − 0.045) (Ω), *M*_B1_(*t*) = *M*_A2_(*t*) = 14887 + 4.929 × 10^5^ (*t* −0.045) (Ω). The weight changes of the two synaptic circuits were Δ|*ω*_1_(*t*)| = −30.615(*t* − 0.045), Δ|*ω*_2_(*t*)| = −30.615(*t* − 0.045), *ω*_1_(*t*) = −0.918 + 30.615(*t* − 0.045), *ω*_2_(*t*) = 0.082 + 30.615(*t*−0.045). The sum of the weighted signals *V*_O_ and output voltage *V*_out_ of each neuron circuit at each time are *V*_O_ = −4.18 + 306.15(*t*−0.045), *V*_out_ = 0.1 V (*t* ≥ 58.65ms), 0 V (*t* < 58.65 ms).

Converging with Figure 3, the theoretical investigation is completely compatible with the simulation outcomes that establishes the correct actualization and analysis of the proposed neuron circuit. However, in [14,15,16,17,18,19,20,21], the relationship between the acting time of the input signal and the weight change in the synaptic circuit was not given, which is unfavorable for the training and further research on MNN circuits.

## 4. Circuit Implementation of the Character Recognition Network

On the basis of the above circuits, a neural network circuit based on the memristors was designed to realize the character picture recognition, which can be extended to recognize any group of characters. As shown in Figure 4a, this work utilized three groups of character pictures (*z*, *v*, and *n*) with the resolution of 3 × 3 as datasets for the ease of simplicity. Each group of pictures includes three standard pictures and 27 noisy pictures. When the circuit training is finished, any *z*, *v*, or *n* character picture inputted into the recognition network circuit in a specific order will be correctly judged by measuring whether the output of the neuron circuit is at a high level.

The three-character recognition network circuit in this paper is composed of three subcircuits with the same structure: Subcircuit *z*, Subcircuit *v*, and Subcircuit *n*. This circuit can also be extended to recognize multiple characters. Different target vectors should be set in the circuit training stage to realize the recognition of each input character. The neuron circuit of Subcircuit *z* is taken as an example in Figure 5 to exhibit the specific working process of the circuit.

Because the goal of this paper was to identify a character image with a resolution of 3 × 3, it was designed to map the pixels in the image to be identified to a one-dimensional vector *In* = [IN_1_ IN_2_ IN_3_ IN_4_ … IN_9_]^T^ according to the sequence shown in Figure 4b and then input it into the nine-input memristive neuron circuit. In this process, logic “1” = 1 V and logic “0” = 0 V were used to represent the black and white in each pixel to realize the picture recognition.

As shown in Figure 5, when certain data in Figure 4a are input into the character recognition network Subcircuit *z*, the mapped input signal *In* and the synapse weight *ω_z_* set in the subcircuit are subjected to matrix multiplication operation to obtain the output voltage *V*_O*z*_ in Figure 5, and then the synapse weight is corrected through further training. Lastly, the trained output voltage is sent to the activation function circuit to obtain the output *V_z_* of the subnetwork.
*V*_O*z*_(*n*) = *ω_z_*× *In*(*n*)(19)

Specifically, the Widrow–Hoff algorithm is used to train the character recognition network circuit. The Widrow–Hoff learning algorithm is an approximate steepest descent method, which uses the mean square error (MSE) as the loss function. Therefore, it is necessary to set the expected outputs of the three subcircuits to *t_z_*, *t_v_*, and *t_n_*, respectively, and then calculate the mean square error with the actual outputs of each subcircuit. Specifically, when the picture *z* is input to the circuit, [*t_z_*, *t_v_*, *t_n_*] = [1, 0, 0] should be set in the algorithm; that is, the expected output of Subcircuit *z* is set to 1, and the expected outputs of Subcircuit v and Subcircuit n are both set to 0. When the input pictures are *v* and *n*, [*t_z_*, *t_v_*, *t_n_*] = [0, 1, 0] and [*t_z_*, *t_v_*, *t_n_*] = [0, 0, 1] should be selected. Taking Subcircuit *z* as an example, the error signal in the *n* iteration is
*e_z_*(*n*) = *t_z_* − *V*_O*z*_(*n*)(20)

Then, the loss function can be obtained as follows:(21)$$\theta_{z}(n)=e_{z}^{2}(n)$$

Because neural network learning aims to find a suitable *ω_z_*(*n*), the mean square error *θ_z_*(*n*) is minimum. Therefore, by using *θ_z_*(*n*), the partial derivative of *ω_z_*(*n*) is calculated, and, after equating the partial result to zero, the minimum value of *θ_z_*(*n*) is obtained. The specific gradient vector equation is as follows:(22)$$\frac{\partial \theta_{z}(n)}{\partial \omega_{z}(n)}=-2e_{z}(n)\times In^{T}(n)$$

Finally, the update amount of weight correction is obtained as shown in the following equation:
Δ*ω_z_*(*n*)
= 2*αe*_*z*_(*n*) × *In*^T^(*n*)(23)
where Δ*ω_z_*(*n*) represents the synaptic circuit weights that need to be updated in the *n* iteration, and *α* is the learning rate. The algorithm will be more accurate if the value of *α* is smaller, but it leads to a slower convergence speed of the algorithm. Therefore, *α* is selected as 0.1. The calculation process of the Widrow–Hoff algorithm can be realized either using a full circuit [28,31,32] or using a combination of software and hardware [33,34,35]. In this paper, Matlab software was used to complete the above iterative calculation.

According to the above method, the standard picture of the *z* character was mapped firstly according to the sequence shown in Figure 4b, and one-dimensional vector *In_z_* = [1 1 0 0 1 0 0 1 1]^T^ could be obtained. The output terminal *V*_O*z*_ of the neuron circuit was connected to the gate of the NMOS transistor; finally, the output *V_z_* could be obtained through this activation function. Because the expected outputs of the character recognition network circuits Subcircuit *z*, Subcircuit *v*, and Subcircuit *n* were different for different input characters, the corresponding correct subcircuit output voltage was 0.1 V, i.e., logic “1”, and the output voltages of the other two subcircuits were all logic “0” = 0 V, so that it could be correctly judged which of the input characters was “*z*”, “*v*”, and “*n*”.

The process of realizing the Widrow–Hoff algorithm by combining software and hardware was as follows: after the circuit output *V*_O*z*_(*n*) was obtained in each iteration, the synaptic weight variation Δ*ω_z_*(*n*) was obtained by external training (Matlab), and the synaptic weight of the circuit was adjusted synchronously. The specific steps were as follows:

*Step I:* Initialization: Set the initial weights of all synaptic circuits in Subcircuit z to zero, the learning rate *α* = 0.1, the maximum training times MAX = 50, [*t_z_*, *t_v_*, *t_n_*] = [1, 0, 0], and the mean square error *θ_z_*(*n*) < 0.005 as the judgment condition of network training termination.

*Step II:* The *n*-th output *V*_O*z*_(*n*) of Subcircuit *z* is taken out and written into Matlab, the error *e_z_*(*n*) is calculated, and then the mean square error *θ_z_*(*n*) is obtained to determine whether to stop the training. Calculate the gradient value of *θ_z_*(*n*) and the weight correction Δ*ω_z_*(*n*) corresponding to the input vector.

*Step III:* Calculate the weight of each synaptic circuit in Subcircuit *z*, the action time of programming voltage, and combine the positive and negative control voltage to correct the weight of each synaptic circuit in subcircuit *z*. At this time, the input voltage of each synaptic circuit in each subcircuit is zero, and the control voltage *V*_G_ remains unchanged.

In this paper, the character recognition network circuit shown in Figure 5 was simulated using LTSpice software. During each training, by observing whether the output of each subcircuit of the character recognition network was correct, the number of incorrectly recognized pictures of the recognition network circuit was recorded. After the character recognition network circuit was trained, 30 character picture datasets in Figure 4a were inputted into the circuit in a certain order to verify whether all pictures could be correctly recognized. The relationship between the number of incorrectly recognized pictures of three neural network circuits and the training times was obtained as shown in Figure 6. It can be observed that, due to the increase in training time, the number of incorrectly recognized pictures of each subcircuit gradually showed a downward trend.

After training the three subcircuits, the accuracy of the identification network circuit was verified. It could be deduced using Equation (10) that the weight change of the synaptic circuit at 0.01 ms was 6.25 × 10^−5^ ≈ 0, whose effect on the circuit could be ignored. Therefore, it was decided to sequentially input each verification picture at an interval of 0.01 ms. The verification process was divided into four stages, as shown in Table 3.

The simulation results of the circuit verification stages are shown in Figure 7. The green, blue, and red curves represent the output voltages of the subcircuits *z*, *v*, and *n*, respectively. The letters between dotted lines represent the input character pictures at this timepoint. For example, when the picture *z* was input at 0–0.01 ms, the output *V_z_* of the neural network Subcircuit *z* was high (0.1 V). The output of other subcircuits was at a low level (0 V), indicating that the circuit successfully recognized the character picture *z*. As can be seen from Figure 7, when the picture datasets were input into the recognition network circuit in the above order, the circuit could output a corresponding correct waveform. Therefore, the proposed three-character recognition network circuit could correctly recognize all character pictures after training.

## 5. Conclusions

This paper primarily focused on the application perspective of memristive neural network (MNN) circuits in the direction of character recognition. A new synaptic circuit based on a memristor and CMOS was proposed. On the basis of this synaptic circuit, an MNN circuit based on the Widrow–Hoff algorithm was designed to recognize three kinds of character pictures. The proposed memristive synaptic circuit could only increase or decrease the weight by inputting the digital logic level. Through mathematical derivation, it was observed that the synaptic circuit had good linearity at the weight programming stage. As a function of the structure of the synaptic circuit, positive, zero, and negative weights were realized. Lastly, the proposed character recognition network was simulated on LTSpice, and the accuracy of the circuit was verified. Therefore, the proposed neural network circuit based on memristors is a promising direction for the hardware implementation of ANNs.

## Figures and Tables

**Figure 1 micromachines-13-02074-f001:**
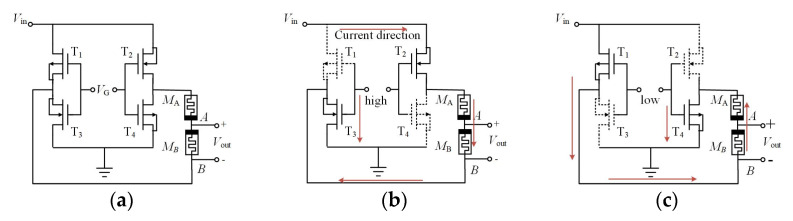
(**a**) Circuit diagram of the memristive synapse circuit; (**b**) current path with positive weight; (**c**) current path with negative weight.

**Figure 2 micromachines-13-02074-f002:**
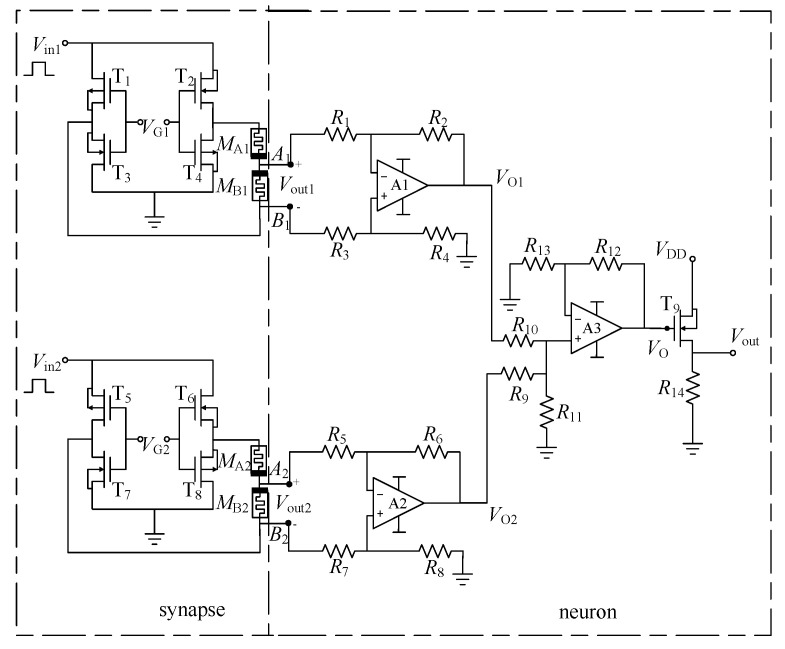
The neuron circuit with two connected synaptic circuits.

**Figure 3 micromachines-13-02074-f003:**
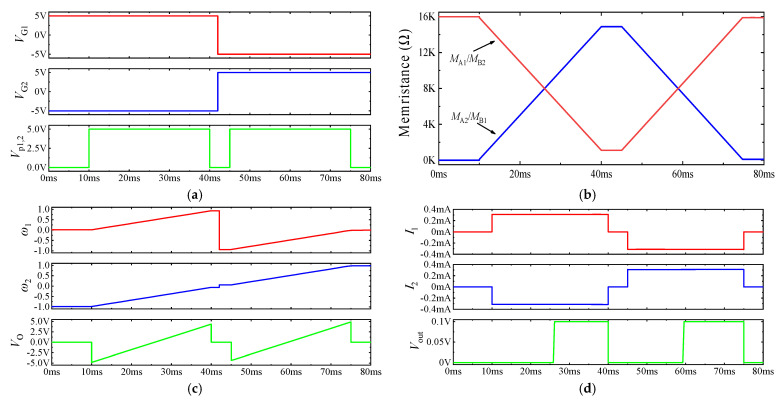
Simulation results of neuron circuit connecting two synaptic circuits: (**a**) control voltage and weight programming voltage; (**b**) resistance curve of the memristor in the synaptic circuit; (**c**) circuit weight and summation voltage *V*_O_; (**d**) current flowing through memristor and output voltage *V*_out_.

**Figure 4 micromachines-13-02074-f004:**
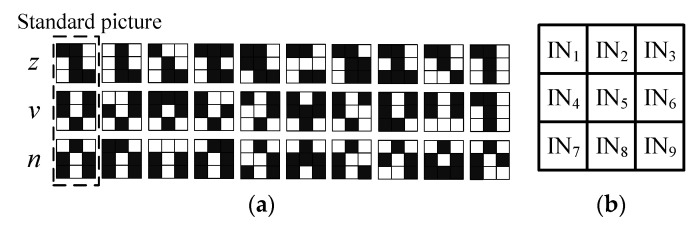
Picture dataset: (**a**) *z*, *v*, and *n* character pictures; (**b**) pixel order of pictures.

**Figure 5 micromachines-13-02074-f005:**
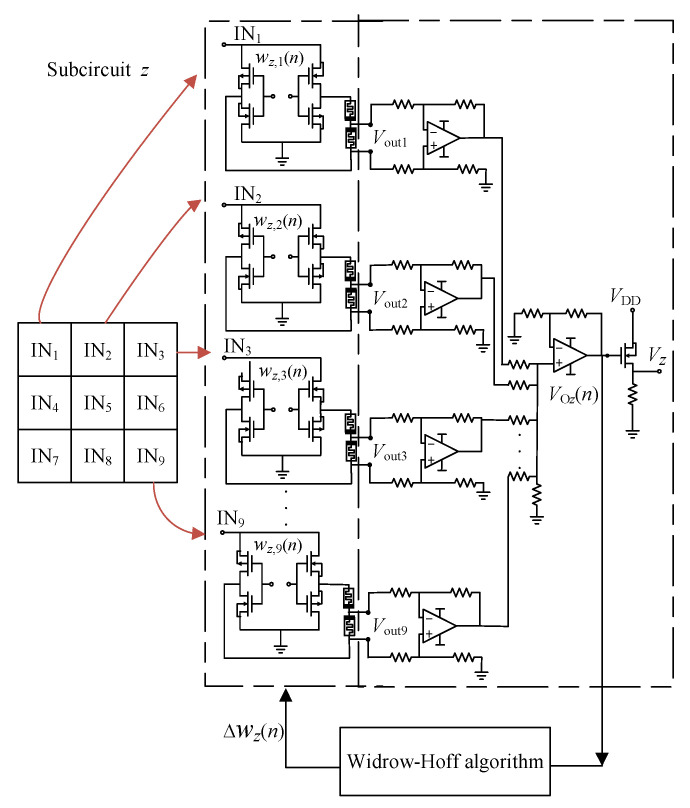
Schematic diagram of neuron circuit of *z*.

**Figure 6 micromachines-13-02074-f006:**
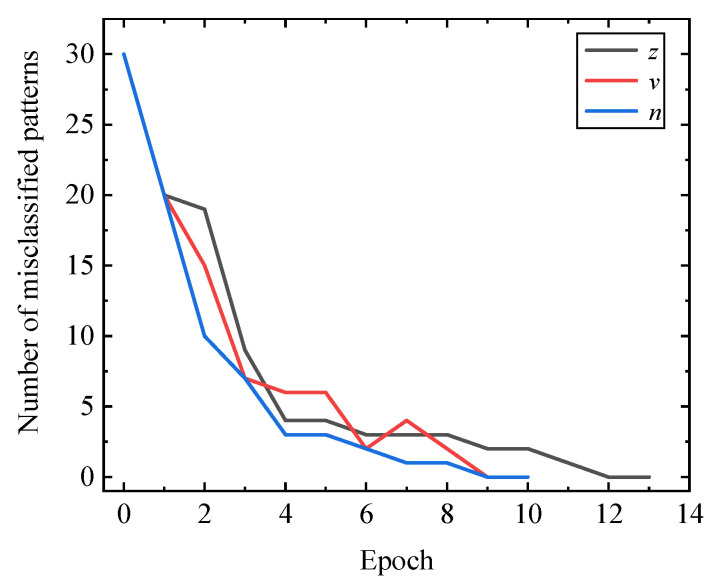
Relationship between the number of misclassifications and training time.

**Figure 7 micromachines-13-02074-f007:**
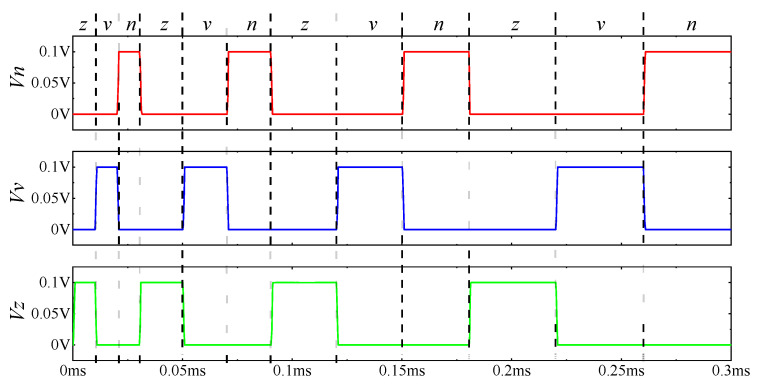
Simulation results of character recognition network circuit and the number of false recognition pictures in the circuit.

**Table 1 micromachines-13-02074-t001:** Conditions w.r.t the control voltage, *V*_G_.

Conditions	Control Voltage	Transistor State
1	*V*_G_ > 0	T_2_, T_3_ = On T_1_, T_4_ = Off
2	*V*_G_ < 0	T_1_, T_4_ = On T_2_, T_3_ = Off

**Table 2 micromachines-13-02074-t002:** Comparison of memristive synaptic circuits.

	[26]	[27]	[28]	[29]	[30]	This Work
Input	Voltage	Voltage	Voltage	Current	Voltage	Voltage
Output	Voltage	Current	Current	Voltage	Current	Voltage
Weight scope	+	+, 0, −	+, 0, −	+, 0, −	+	+, 0, −
Weight linearity	Yes	Yes	No	No	No	Yes
Number of memristors	2	2	1	5	1	2
Number of control voltages	1	2	2	2	1	1

**Table 3 micromachines-13-02074-t003:** Various stages of verification.

Stages	Time (ms)	Observation
I	0–0.03	Input three standard pictures of *z*, *v*, and *n* into the circuit in turn.
II	0.03–0.05	Continuously input the noisy pictures of each character into the circuit twice in turn
III	0.09–0.12	Continuously input the noisy pictures of each character into the circuit three times in turn
IV	0.18–0.3	Continuously input the noisy pictures of each character into the circuit four times in turn

## Data Availability

Not applicable.

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
