# Peer review of "A Novel Memristive Neural Network Circuit and Its Application in Character Recognition"

_micromachines, 2022, doi:10.3390/mi13122074_

Round 1

Reviewer 1 Report

This paper presents a new synaptic circuit based on memristor and CMOS and according to this synaptic circuit, a MNN circuit based on the Widrow-Hoff algorithm is designed to recognize three kinds of character pictures. Furthermore, the accuracy of the circuit is verified. The paper is well organized with good English writing. Some minor notes that may be of use:

1: What’s the specific type of the T1-T8 and A1-A3 in Figure 2 on page 9?

2: I think the figure legend “Time domain waveform diagrams of x, y, and z”of Fig.2 is wrong, It does not match the contents of the graph. PLS check.

3: Some representation of symbol in the paper is expressed in inconsistent ways, like the “z, v, n ” in Figure 4 is differ from the expressions of them in the text. So the authors should maintain consistency in all the expressions in the whole text.

Author Response

PLS see the attachment.

Reviewer 2 Report

The manuscript proposed a memristor-based neural network configuration. The paper is well written. The idea is novel and interesting. The analysis is sufficient. However, I suggest that the authors address some minor corrections.

(1)  There are some minor errors in the text, like the illustration of the figure 2 does not match the content of the figure.

(2)    On page 4, it is wrote as As the resistance of MB is in [100,16k], theoretical analysis shows that the range of â–³t is in [0,0.032]. I think the authors need to further explain the details of obtaining the scope of â–³t in equation (8).

 (3)  The author should add some more references in recent 5 years.

Author Response

PLS see the attachment.
